# Exploring the Potential of Alternate Inorganic Fibers for Automotive Composites

**DOI:** 10.3390/polym14224946

**Published:** 2022-11-16

**Authors:** Muhammad Shoaib, Hafsa Jamshaid, Mubark Alshareef, Fahad Ayesh Alharthi, Mumtaz Ali, Muhammad Waqas

**Affiliations:** 1School of Engineering and Technology, National Textile University, Faisalabad 37640, Pakistan; 2Department of Chemistry, Faculty of Applied Science, Umm Al Qura University, Makkah 24230, Saudi Arabia; 3Desalination Technologies Research Institute, Al Jubail 35418, Saudi Arabia

**Keywords:** fiber reinforced composites, inorganic fibers, rockwool, basalt, thermo-mechanical properties

## Abstract

Composites are a promising material for high-specific strength applications; specifically, fiber-reinforced polymer composites (FRPCs) are in the limelight for their extraordinary mechanical properties. Amongst all FRPCs, carbon fiber reinforcements are dominant in the aerospace and automotive industry; however, their high cost poses a great obstacle in commercial-scale manufacturing. To this end, we explored alternate low-cost inorganic fibers such as basalt and rockwool as potential replacements for carbon fiber composites. In addition to fibrous inclusions to polymers, composites were also fabricated with inclusions of their respective particulates formed using ball milling of fibers. Considering automotive applications, composites’ mechanical and thermo-mechanical properties were compared for all samples. Regarding mechanical properties, rockwool fiber and basalt fiber composites showed 30.95% and 20.77% higher impact strength than carbon fiber, respectively. In addition, rockwool and basalt fiber composites are less stiff than carbon and can be used in low-end applications in the automotive industry. Moreover, rockwool and basalt fiber composites are more thermally stable than carbon fiber. Thermogravimetric analysis of carbon fiber composites showed 10.10 % and 9.98 % higher weight loss than basalt and rockwool fiber composites, respectively. Apart from better impact and thermal properties, the low cost of rockwool and basalt fibers provides a key advantage to these alternate fibers at the commercial scale.

## 1. Introduction

The rapid growth of manufacturing sectors has necessitated the improvement of materials in terms of their specific strength, as well as their affordability. In this regard, composite materials have emerged as one potential material with ideal qualities, making them suitable for a wide range of applications [1,2,3]. Composites have found significance in construction, machinery, automobiles, aircraft, medicine, and transportation [4,5,6]. Due to considerable improvements in structural, mechanical, and tribological characteristics of fiber-reinforced composite (FRC) materials, these are being used extensively in several advanced applications [7,8,9]. Despite FRC’s success in making more durable products, there is reason to be concerned about the cost and performance ratio. Scientists all around the globe are concerned with developing low-cost materials without compromising performance, which can ensure commercialization [10,11,12]. The present work is an extension of this same idea, where we employed rockwool and basalt fibers as an alternative reinforcement of expensive “carbon” fibers.

Carbon-fiber-reinforced polymer composites (CFRPs) are used in high-tech engineering applications due to their superior strength-to-weight ratio [13,14,15]. Due to economic and commercial considerations, there is a less stringent demand for performance, whereas a higher concern is manufacturing rate. Therefore, scientists found a replacement to resolve the cost and recycling issue, and materials had comparable properties to CFRPs. One such material of interest that is now widely used is basalt fiber, basalt fiber is made from black, fine-grained volcanic rock called Basalt. Basalt is a hard, thick, inert igneous rock that melts when heated like solidified lava. Basalt rocks are crushed to fractions of 5–40 mm, separated of metal and magnetic impurities by magnetic separation, screened, and washed of minute inclusions (dust, etc.), and then dried either by natural air circulation or in a special dryer. The raw material is regularly put into the loader above the smelter. BF is made by melting basalt at 1450 °C. Molten rock is extruded via tiny nozzles to make BF filaments, which are cost-effective and superior to glass fibers [13,14,16,17]. They are non-toxic and non-carcinogenic, making them better for the environment and human health. They are resistant to heat, chemicals, and have a low absorption rate [18].

Another quality alternative is rockwool—or stone wool—which is also a natural inorganic fiber with less environmental impact. Rockwool fibers were synthesized at high temperatures using raw materials like basalt, diabase, limestone, and dolomite, which are combined and fed into a furnace where the melt is prepared and passed through spinnerets to obtain filaments [19]. It can be used primarily for thermal and acoustical insulation, typically in buildings, vehicles, and other industrial equipment. It has suitable tensile strength, chemical resistance, dimensional stability, and excellent insulation properties [20,21]. Compared to glass-fiber-reinforced plastics (GFRP), basalt-fiber-reinforced plastics (BFRP) were shown to be better in terms of compressive strength, Young’s modulus, and flexural behavior [22,23]. Manikandan et al. also reported similar results, where basalt fiber was used as a potential replacement for GFRP [24,25]. To improve its performance, surface modification of basalt fibers was carried out [24]. Basalt has been widely used as an exterior or internal reinforcement inside concrete materials, having first gained popularity in the building sector as fibers [26,27,28]. In addition to its usage in marine [29], impact [24,30], and ballistic defense [31,32], basalt has applications in thermal insulation. By dicing up the basalt fiber and blending polypropylene and clay, Eslami-Farsani et al. fabricated a composite with enhanced yield strength and elastic modulus [33]. According to a study by Manikandan et al., the performance of stone wool composites was superior to that of glass-fiber-reinforced composites [34]. Another study also presented the high performance of stone wool composites in terms of young modulus, impact force, and compressive and bending strength. These good properties enable the application of stone wool fibers, which could be attractive for fiber-reinforced composites [35]. Bredikhin, P. A. et al. stated that the addition of 40 wt.% of waste rockwool to polyethylene raises the material’s bending stress by 20%, impact energy by 40%, and hardness by 50%. The performance of composites made with pristine rockwool fillers was superior to that of composites made with recycled rockwool [36]. Paivo Kinnunen et al. found that the compressive strength of geopolymers made using the newly disclosed process was more than 12 MPa, and they included 33% rockwool and 47% fly ash [37]. Such intriguing applications of rockwool fibers are due to their low density, excellent heat resistance, sound insulation, and being sustainable and environmentally friendly [38]. In other work in the literature, 20 wt.% rockwool polymer composites showed a significant increase in tensile strength, yield strength, and tensile modulus; 20 wt.% rockwool fiber provides suitable fiber dispersion in the matrix and improves the load transfer; 20 wt.% achieves excellent tensile properties due to strong interfacial binding, absence of vacancy, and strong matrix–fiber adhesion. A total of 30% of the composites showed excessive micro voids fractures and failure at low strain. Due to the fast propagation of fracture and low deformation, 30 wt.% of the sample exhibits lower tensile strength. A finite element model (FEM) was investigated under comparable boundary conditions. Experimental and FEM analysis showed similar tensile modulus trends. Both models were 90% accurate, demonstrating high agreement. The robust model accurately predicts rockwool polymer composites behavior. As a polymer composite reinforcement, rockwool fiber improves mechanical properties, particularly tensile [39]. Compared to other textile fibers, rockwool is a recent specie of fiber reinforcements and its mechanical properties are less understood. Due to rising environmental norms and awareness, there is a growing interest in replacing natural raw materials with synthetic materials that deposit in landfills and cause pollution. Cost-effective, non-toxic, and non-carcinogenic behaviors trigger the interest in such materials. However, studies related to the application of rockwool in polymer composites are missing in the literature.

Therefore, for the first time, we compared the thermo-mechanical properties of rockwool composites with carbon- and basalt-fiber-reinforced polymer composites. Bearing automotive applications in mind, we studied the thermo-mechanical properties of the composites. Rockwool showed low tensile strength; however, suitably high impact and thermal properties were recorded. In addition to fiber reinforcements, we also incorporated ball-milled particles of the fibers in composites.

## 2. Materials and Methods

### 2.1. Materials

Short dry waste carbon fibers of carbon were sourced from the local industry. Rockwool fibers were sourced from the M/S Incom Rockwool (Pvt) Ltd. Hattar, Pakistan. Basalt fibers were procured from Kamenny Vek corporation, Moscow, Russia. Epoxy resin (EPIKOTE 816) based on diglycidyl ether of bisphenol A (DGEBA) containing a proportion of reactive diluent, Cardura E10P (glycidyl ester of neodecanoic acid), and hardener (EPIKURE F205) formulated from isophorone diamine (IPDA) were sourced from HEXION INC. Columbus, OH, USA.

The physical properties of fibers are important to consider, as the final properties of the composites are strongly dependent on the intrinsic properties of the fibers. Considering this, the properties of fibers are summarized in Table 1.

### 2.2. Milling of Fibers

The average size of fibers was calculated to be 52 mm for dry waste carbon, 20.8 mm for rockwool, and 43.8 mm for basalt fibers. All the fibers were carded and opened to obtain a fluffy look. Afterwards, all fibers were made moisture-free by oven drying for 24 h at 100 °C. Dried fibers were ball milled for a 30 min cycle by using steel balls at a speed of 300 rpm. The fine powder was obtained by milling each fiber. A zeta sizer was used to calculate the sizes of the particulates. Following are the measured sizes of the particulates of the fibers: carbon fiber particulates 608 nm, basalt fiber particulates 1158 nm, and rockwool fiber particulates 1228 nm.

### 2.3. Fabrication of the Composites

The fiber-reinforced and fiber-particulates-reinforced composite laminates were fabricated by using the hand layup method followed by a low-compression molding technique, as shown in Figure 1. The resin and hardener were mixed at a ratio of 5:3 after optimization, before application on reinforcements. Reinforcement was taken at 5% of the total weight of matrix material and particulates were taken at 1.5% of the total weight of matrix material. To avoid the sticking of resin, firstly the silica release gel was employed on the surface of the glass mold. The fiber was mixed in epoxy resin and poured into the mold. After that, a metal roller was gently rolled over it to remove air bubbles and trapped gasses. After proper application of resin, wet laminates were compressed by tightening the mold nuts and given low compression. Composites were laid for 24 h at room temperature and were post-cured for 1 h at 100 °C temperature.

All composites were coded according to Table 2, mentioned below.

### 2.4. Characterizations

#### 2.4.1. Mechanical Characterization

Different static mechanical tests, i.e., tensile, flexural, and Charpy impact tests were performed on composite samples to compare their mechanical properties. Tensile properties of composites were investigated according to ASTM D3039, with a gauge length of 120 mm and at 2 mm/min crosshead speed using the 100 KN load cell, and 200 mm long and 25 mm wide samples were used for tensile testing. Flexural (3-point) properties were performed at a 1 mm/min loading rate, using an 80 mm span length with a load cell of 20 KN according to ASTM D7264 standard test method. For flexural testing, 120 mm long and 13 mm wide samples were used. Both tensile and flexural properties were investigated on the universal testing machine (Zwick Roell, Z100, Ulm, Germany). To investigate the in-plane impact properties of laminates, a pendulum impact tester (Zwick/Roell, HIT50P, Ulm, Germany) with a Charpy instrumented hammer with 25 J energy was used by following the ISO-179 standard test method; 100 mm long and 10 mm wide samples were used for the Charpy impact testing. Five repetitions for each mechanical test were performed and the average value was reported. Dynamic mechanical analysis (DMA) is a method often used to describe the characteristics of a material as a function of temperature, time, frequency, stress, environment, or a combination of these factors. Dynamic mechanical analysis (DMA) conditions were set as frequency 5 Hz, temperature 25–120 °C, heating rate 3 °C/min, and testing mode of 3-point bending.

#### 2.4.2. Thermal Characterization

Thermogravimetric analysis (TGA) measures the weight change that takes place while a sample is heated at a consistent pace and may be used to ascertain a material’s thermal stability and the percentage of volatile components. TGA analysis proceeded from 25 to 600 °C with a heating rate of 5 °C/min under N_2_ atmosphere. Tests were performed using STA 8000 Perkin Elmer (Waltham, MA, USA) TGA analyzer. Material’s coefficient of thermal expansion (CTE) CTE is an indirect measure of a material’s increase in dimensions under heating. This coefficient is calculated under conditions where the sample is held at constant pressure and is not anticipated to undergo a phase transition. This proceeded from 25 °C to 150 °C, under a heating rate of 5 °C/min, using DIL 801 L TA (Waltham, MA, USA) dilatometer for thermal expansion.

#### 2.4.3. Morphological Analysis

Scanning electron microscopy (SEM) was carried out for the composites after mechanical (tensile) testing. The samples for SEM (MIRA 3, TESCAN, Brno, The Czech Republic) were sputtered with gold in argon gas atmosphere, which enables high-quality imaging. The thickness of gold plating was kept at 2 nm using a current of 20 mA. The samples were visualized in a nitrogen atmosphere with SE (secondary electron) detector, using an acceleration voltage of 10 kV. The working distance was maintained at 16–32 mm with scan mode.

## 3. Results and Discussion

### 3.1. Mechanical Testing

Composites used in automotive applications are commonly prone to different types of loads, therefore their mechanical properties are crucial in determining their suitability in such applications. Here, we tested the mechanical properties such as tensile strength, flexural strength, and impact strength. These mechanical responses were checked for fiber reinforcement and fiber with their particulate reinforcements, as mentioned in Table 1. Three repetitions for each mechanical test of different composite laminates were performed, and average values of results were reported.

#### 3.1.1. Tensile Testing

Tensile characteristics are a measure of a material’s ability to withstand tensional pressures. The composites’ tensile properties indicate the materials’ deformation behavior under tension mode. Figure 2a shows the stress–strain behavior of the composites. Characteristically, reinforcement depicts the mechanical properties of the composites. Amongst all, the CFP composite bore the maximum stress linearly because of the superior properties of the carbon fiber along with the inclusion of its particulates. Composite BFP showed a nearly similar stress–strain response for the sample containing particle inclusion of the basalt fiber. CF and BFP composites had comparable results, as the inclusions of the basalt particulates play a critical role in increasing the strength of the composite; as fibers show excellent properties only in one direction, the inclusion of the particulates increased the properties in all directions [42]. BF had slightly lower results as it can sustain low stress, i.e., basalt fiber has 26% lower tensile strength than carbon fiber. On the other hand, rockwool-reinforced composites showed the least result in tensile behavior due to their very low strength as compared to basalt and carbon fibers. Even the inclusion of the particulates of the rockwool is lower in the stress-bearing capacity of the composites due to improper adhesion of the fiber and particulates with the matrix material. On the other hand, rockwool-reinforced composites showed the least result in tensile behavior due to their very low strength as compared to basalt and carbon fibers [41]. Experimental observations by Liu and Wilbrink et al. revealed that bond failure between fibers and frictional sliding also significantly impact deformation and damage of rockwool [43,44].

Breaking strength or ultimate strength is the maximum stress that a material can bear before breaking under tensile loading. Figure 2b shows the tensile strength trend of the composites. The inclusion of the particulates increased tensile strength in CF and BF. In CFP the inclusion of carbon particulates’ tensile modulus increased by 37.88% from CF composite and 15.88% in BFP tensile modulus from BF composites. It was found that the inclusion of particles influences Young’s modulus of the composite and Young’s modulus increases as the size of particles decrease at the nanoscale because nanoscale particle sizes require higher stress levels to cause noticeable debonding, and because of the greater reactive surface area per unit volume of the particles that developed strong interaction with the matrix [45]. Unlike other samples, the RFP composite showed a decrease in ultimate strength compared to RF. This could be related to poor dispersion and the weak interface of rockwool fibers particles. as rockwool is an impure form of basalt, formed by the addition of clay to the basalt; this clay impurity results in a brittle nature and lower tensile strength in rockwool fibers.

The tensile modulus is a direct measure of a material’s stiffness; specifically, it is a function of the initial extension of material when subjected to a particular load. The higher a material’s tensile modulus is, the more force is required to deform it. Figure 2c shows the tensile modulus of the composites. It can be seen, for CFP, that the maximum stiffness and particulate reinforcement increased the tensile modulus by 35%, as compared to CF. While basalt and rockwool composites showed less stiffness than carbon fiber composites due to higher initial extension, BFP showed better stiffness than BF due to particulate inclusion which enhanced the mechanical characteristics of the composites. RFP and RF showed comparable results in their stiffness, whereas RFP showed nearly no difference in modulus, even after particulate inclusions.

#### 3.1.2. Flexural Testing

The resistance to material bending is approximated from flexural testing, which involves bending a polymeric beam under a specified force. The flexural modulus of a material is a measure of how far it can be bent without being permanently damaged. Figure 3a shows the flexural behavior of the composites. Carbon fiber and the inclusion of their particulate-reinforced composites showed comparatively better results. CFP showed better flexural properties, making the composite stiffer and more rigid, as carbon has superior qualities to basalt and rockwool-based composites because of its intrinsic compact fiber structure. It can also be observed that basalt has intermediate bending resistance and rockwool showed the least. The span-to-depth ratio for the flexural test was calculated using Equation (1):Span length = Thickness (mm) ∗ 32(1)

Figure 3b shows a similar trend as particulate-inclusion composites have better resistance than only fiber-based composites, which is due to better stress distribution over the matrix. The addition of particles increases the fiber fraction and increases the crystallinity, thus enabling higher resistance to bending. Particulate reinforcements inhibit the crack propagation in the matrix at a much smaller scale and in multiple dimensions, therefore enhanced properties were observed in particulate inclusions. Rockwool fibers contain clay impurities, which add free volume or amorphous domains in fibers, hence larger deformation is possible in rockwool fibers.

#### 3.1.3. Impact Testing

Impact strength, also called impact toughness, is the amount of energy that a material can withstand under sudden load. It may also be defined as the threshold of force per unit area before the material undergoes fracture [46]. Figure 4 shows the impact strength behavior of the composites. Tough materials have a combination of strength and ductility, as these are essential requirements for higher impact resistance. On the other hand, higher stiffness or brittleness is related to lower impact toughness, compared to flexible materials. In these scenarios, particulates increased stiffness and showed low-impact strength. Particulates inclusion enhanced the stiffness of the composites so they showed low impact strength [47]. The addition of particles causes a reduction in impact resistance, due to stresses concentrating around particles and causing the cracks to form around the filler and spread. Figure 4 shows that RF composite absorbed higher impact energy and toughness, i.e., 42% higher than CF composite and 11% from BF composite. Lower impact strength of BF and CF is related to higher stiffness; therefore, CF composite had the lowest impact strength. BF and BFP composites had comparable results due to fewer differences in the stiffness of both composites. With 34% of basalt, impact energy absorption rises to 58%. Carbon fiber increases brittleness, whereas basalt fiber makes hybrid composites more ductile. Carbon fiber composites are prone to stress concentration and impact damage. Fiber fracture uses less energy than fiber pullout. The shear force on fibers may surpass the fiber/matrix load and debonding may occur. When stress surpasses fiber strength, it fractures. Owing to the excellent interaction between basalt fibers and the phenolic matrix, the hybrid composites’ energy absorption due to fiber pull-out increases [48]. Comparing the results with and without particulate inclusion, it can be observed that without particles the impact resistance is higher. Specifically, the rockwool particles lowered the impact strength most drastically, which is due to their relatively small size and high surface area.

#### 3.1.4. DMA Analysis

Dynamic mechanical analysis (DMA) is a technique used to investigate the solid-state rheology of composite materials. It also provides an approximation of the viscoelastic properties of polymeric materials. The strain in the material is measured as it is subjected to sinusoidal tension and complex modulus of the material is determined. Changing the sample temperature or the stress frequency is commonly employed to manipulate the complex modulus. This technique is used to find transitions associated with polymer chain motions and identification of the glass transition temperature of the material. Indicative of the stiffness of composite material, the viscoelastic storage modulus also shows the energy stored in a sample subjected to sinusoidal strain. It shows the relation between storage modulus and temperature, which may be used to infer properties including the material’s stiffness, the degree of cross-linking, and fiber/matrix interfacial bonding [49]. A material’s degree of structure may be inferred from its storage modulus. It represents the elastic potential energy of the sample. The Tan delta is the ratio of the loss modulus to the storage modulus, and the loss modulus represents the viscous component of the sample, or the amount of energy lost.

Figure 5 displays the results of the composites’ DMA analyses. Figure 5a depicts the storage modulus behavior of composites. Because of the extraordinary properties of carbon fiber, carbon fiber composites demonstrated better performance. The incorporation of particles improved the composites’ mechanical properties; hence, CFP composites are superior to CF composites. The storage modulus explains the elastic region of the composites, and carbon bore the highest elastic stress compared to others. The storage modulus of BFP and BF composites is equivalent; however, the storage modulus of RFP is greater than RF composites.

Figure 5b depicts the loss modulus of the composites, which is the measure of the energy dissipated in the form of heat when the material turns viscous. The loss modulus of the CFP composites was higher, because of the inhibition of the chain relaxation process within the composite. While particulate inclusion enhanced the loss modulus values of the composites, Figure 5b shows that for all composites, inclusion of their particulates had a higher value of the loss modulus than their fiber-reinforced composites. Loss modulus represents the capacity of energy dissipation after the addition of particles in the composite. The fibrous particles allow multidimensional energy dissipation at the micro-level, therefore an increase in loss modulus is observed. Compared to other fibers, poor adhesion of rockwool with the matrix was not effective in the increase of loss modulus [50]. After CFP the BFP, CF, BF, RFP, and RF composites, respectively showed the loss modulus value. Khan et al. indicated that the addition of multi-walled CNT particles considerably improved the damping property measured as a rise in the loss modulus of carbon-fiber-reinforced polymer composites; 3 wt% of the particles improved tensile strength by 20% and flexural strength by 35.7%. The interaction of particles with epoxy enhances interfacial stress, which had a substantial influence on natural frequency and damping ratio [51]. The tan delta curves for all composites are shown in Figure 5c. With the incorporation of particulates, the tan delta peak is higher due to less restriction in the movement of the polymer molecules caused by the presence of less stiff fibers. However, when the particulate content is low, there will be regions with high concentrations of resin (matrix) in the composite which will not be affected by the presence of particulates. All Tg values for the composites were within 50–65 °C in fact, in the same range of Tg for the pure resin (70–75 °C). Thus, the other fibers showed more influence on the tan delta peak than in the measured glass transition temperature. The increase in fiber content showed a decrease in the tan delta peak because the overall interface area within the composite was increased by the fibers.

### 3.2. Thermal Testing

Thermal techniques can provide information for material development and selection process optimization, engineering design, and prediction of end-use performance. We performed thermogravimetric analysis (TGA) and coefficient of thermal expansion (CTE) to evaluate the different composites. Additionally, for automotive applications, there is a constant exposure to heat, therefore the thermal properties of composites are important to consider.

#### 3.2.1. Thermogravimetric Analysis (TGA)

It is important to determine the thermal properties of composites before they begin to show signs of wear and tear. The TGA was used to determine the weight loss of the composite as a function of increasing temperature. To assess the thermal stability of composites, the TGA study was carried out, which was followed by the presentation of a comparative analysis between various kinds of composites, as shown in Figure 6. The thermogram shows a progressive weight loss as the temperature is raised, with the weight loss beginning at around 330–360 °C. It was also observed that quantitative chain rupture causes a significant deterioration phase in the epoxy matrix of the composites, at temperatures between 330 and 370 °C. Our findings are identical to previous reports, where the highest degradation occurs in the temperature range of 320–380 °C for epoxy composites. Composites degrade in two stages: first by becoming brittle, and then by crumbling [52]. The first decomposition occurs in the temperature range of 320 to 390 °C, which corresponds to the decomposition of the epoxy matrix. Degradation temperature for inorganic fiber-based composites is often found between the decomposition temperature of the reinforcement and the polymer matrix. The temperature ranged from 410 to 530 °C in the second stage, which corresponds to the decomposition of fibers. Because of the excellent thermal stability of the basalt fiber and rockwool, their composites showed a lower weight loss when subjected to temperature increase. Comparatively, CF composite had a higher mass loss because of the high-temperature conductivity of CF, which conducts the heat in the composite sample, resulting in higher degradation. On the other hand, RF and RFP composite has higher insulation due to the non-conductive nature of the sample. Because of the better thermal stability of these inorganic natural fibers, their application in automotive applications poses huge potential, and, more specifically, they may be better than cellulosic fibers. Thermal stability is critical for applications like “engine cover”, where the average temperature stays at 50 °C. Based on lower thermal degradation, we expect longer life for automotive parts made from RF and BF composites [53].

#### 3.2.2. Coefficient of Thermal Expansion (CTE)

Thermal expansion is the property of a material that increases its volume or alters its shape because of an increase in heat. The average kinetic energy of the molecules increases as the temperature rises; hence, the frequency of molecular vibration also increases. The coefficient of thermal expansion is defined as the ratio of the relative expansion of materials per unit increase in temperature. Figure 7 shows the expansion in the temperature range of 50–150 °C; it demonstrates that the inclusion of particulate reinforcement showed relatively higher expansion, compared to without particles. CFP composite demonstrated the most expansion compared to other composites, as carbon is thermally conductive, and expansion occurred by absorbance of heat. Other composites reinforcements are thermally stable, so showed less expansion on heating the composites. In fiber-reinforced composites, RF showed the least expansion, followed by BF composites, which can be attributed to their lower conductivity. Lower thermal expansion relates to lower shape distortion under heat, which is an important feature to avoid temperature-induced dimensional changes in automotive parts. In this regard, the lower thermal expansion of RF composites is an interesting feature.

### 3.3. Morphological Analysis of Fracture Surfaces

From the SEM images, it can be observed that all the composites showed fiber rupture, as well as fiber pull-out. At lower resolution (top row), the fiber diameter of basalt, rockwool, and carbon can be observed clearly in Figure 8a–c, respectively. Amongst all fibers, carbon fibers (Figure 8c) were the finest and offered a stronger interface with resin. In contrast, the coarser fibers of basalt and rockwool can be seen in the SEM (Figure 8a,b). It is important to note that rockwool fibers showed a greater variation in diameter, i.e., highly finer fibers were also observed at higher resolution (Figure 8b-center row). Additionally, there was difference in the interface according to difference in fiber diameter; for example, stronger fiber interface was observed for finer fibers, as compared to coarser ones. In contrast, a weaker fiber-resin interface can be observed for the basalt fibers, as highlighted in Figure 8b, center row). The fiber pull-out was prominent in BFP and RFP, which is related to the relatively poor interface of these fibers with the matrix. Particulates of CFP were also completely dispersed, without any aggregate formation. It is important to note that the carbon particles had porous interface, which can be completely infused with resin, thus offering a strong interface. An overlapped intimate interface of carbon fiber particles and polymer resin is highlighted in Figure 8c, bottom row. Rockwool fibers come second to carbon particles regarding interface strength, which also has a slightly porous structure (Figure 8b-bottom row). Extremely small pores of such particles are related to impurities added during processing. In contrast, the solid structure and poor compatibility of basalt particles with resin render poor interface, as highlighted in Figure 8a-bottom row.

## 4. Conclusions and Future Work

In this work, fiber-reinforced and associated particulate-reinforced composites were fabricated and their mechano-thermal characterizations were carried out. Compared to conventionally used carbon fiber reinforcement, the cost of basalt and rockwool is four and ten times lower, respectively. Considering the low-cost alternative of rockwool, this study compared the thermo-mechanical properties of rockwool fibers. The effect of fiber and associated fiber particulates on the mechanical properties was studied on their on their tensile, flexural, Charpy impact, and dynamic mechanical properties. Thermogravimetric analysis (TGA) and coefficient of thermal expansion (CTE) were investigated to approximate the thermal properties. In addition, rockwool fiber and basalt fiber composites demonstrated a 30.95 % and 20.77 % greater impact strength than carbon fiber composites, respectively. In addition, rockwool and basalt fiber composites are less rigid than carbon and are suitable for low-end automobile applications. Additionally, rockwool and basalt fiber composites are more thermally stable than carbon fiber composites. Thermogravimetric examination of carbon fiber composites revealed 10% higher weight loss than both basalt and rockwool fiber composites. In addition to superior impact and thermal qualities, the inexpensive cost of rockwool and basalt fibers on a commercial scale is another significant benefit of these alternative fibers.

This study observed significantly better thermal and impact properties of rockwool fibers, which are major drawbacks of carbon fiber composites. Ideally, such properties are critical for automotive composite applications, which was a major practical area of the study. Blends of carbon and rockwool fibers can be studied in the future, to achieve a synergistic combination of properties. The low cost of rockwool fibers comes with an obstacle of low performance, mainly due to weaker interface. Surface modification of fibers to render a stronger interface could be an important consideration for future work. Particulates of inorganic fibers are effective for resin modification, therefore the effect of ball-milling time and concentration of different fiber particulates derived from fiber waste can also be a part of future work. A great focus has been given to carbon fiber modifications and process optimization in continuous manufacturing, which is still missing for these alternate fibers.

## Figures and Tables

**Figure 1 polymers-14-04946-f001:**
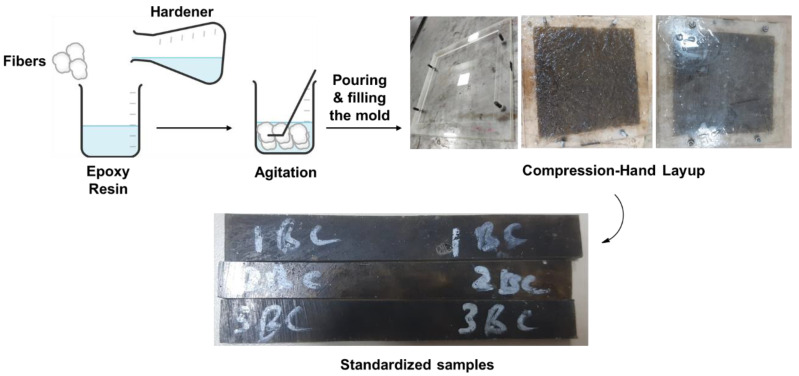
The fabrication process of the composites.

**Figure 2 polymers-14-04946-f002:**
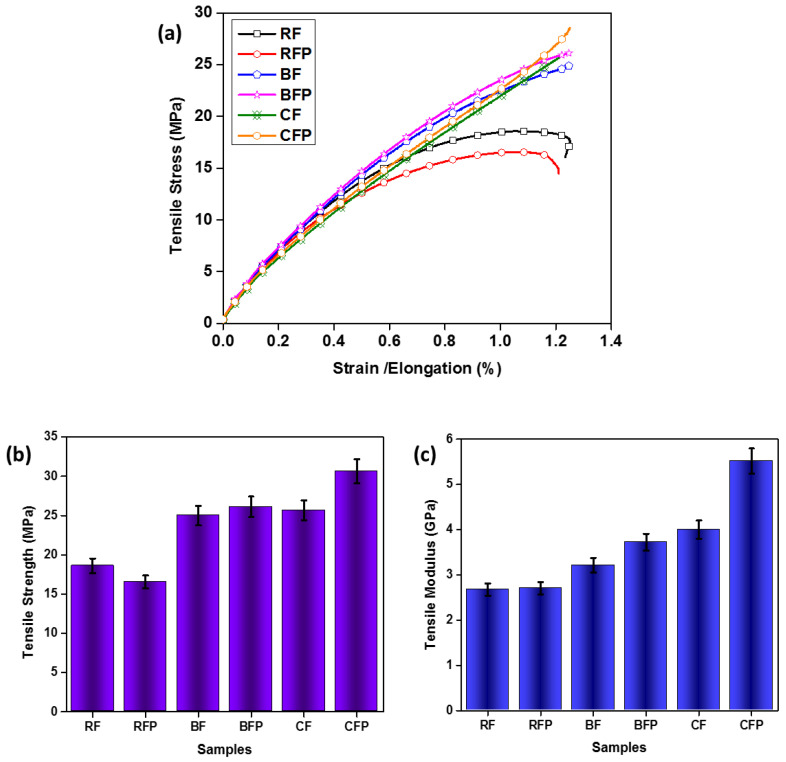
The behavior of the composites in tensile testing (**a**) Stress–strain behavior (**b**) Tensile Strength (**c**) Tensile Modulus.

**Figure 3 polymers-14-04946-f003:**
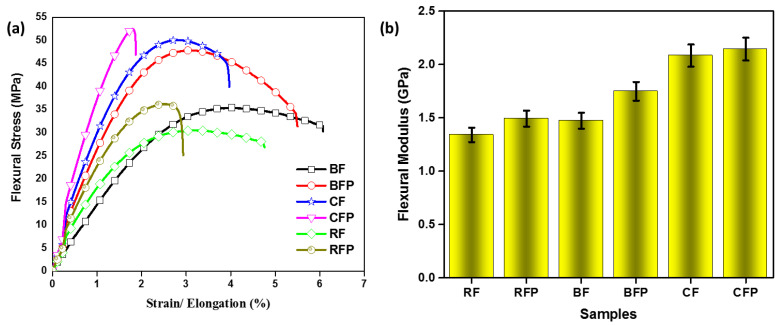
The three-point bending analysis of samples is approximated from (**a**) stress–strain behavior and (**b**) flexural modulus.

**Figure 4 polymers-14-04946-f004:**
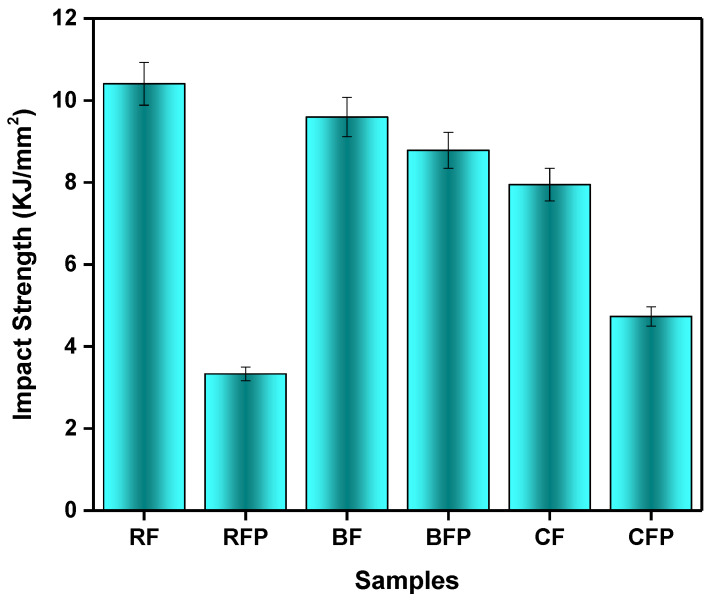
Impact strength comparison of the composites.

**Figure 5 polymers-14-04946-f005:**
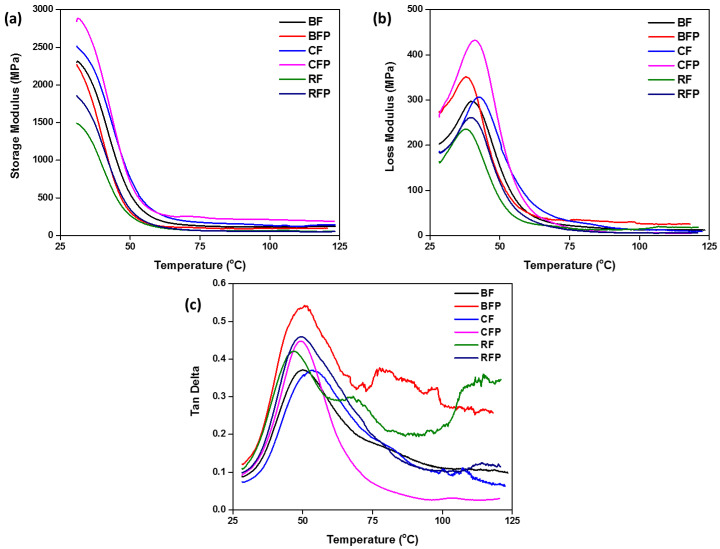
The comparison of the composites’ dynamic mechanical analysis shows (**a**) storage modulus, (**b**) loss modulus, and (**c**) Tan Delta.

**Figure 6 polymers-14-04946-f006:**
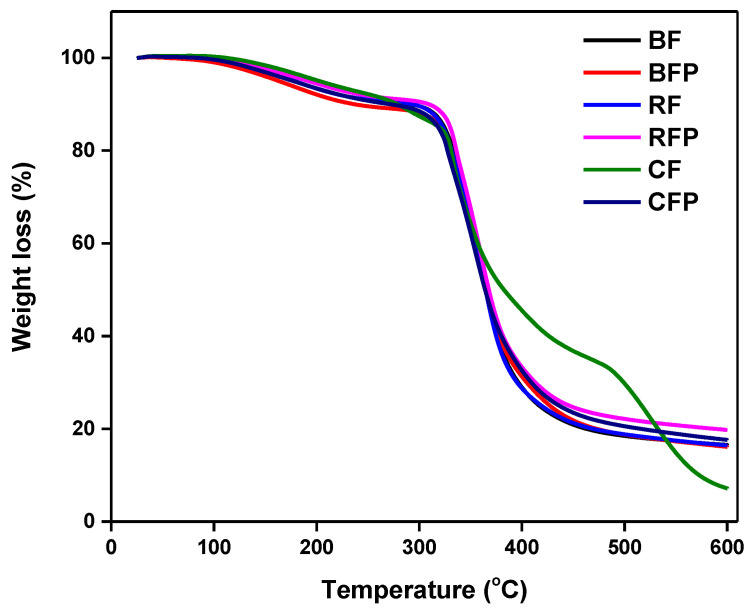
TGA analysis of the composites.

**Figure 7 polymers-14-04946-f007:**
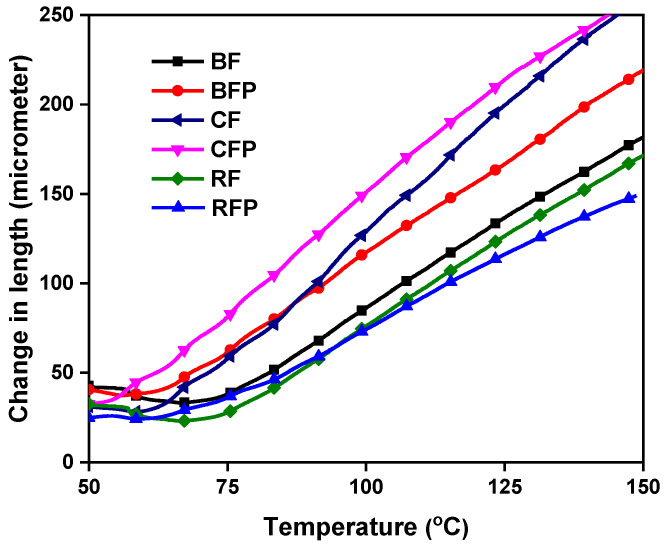
Thermal expansion behavior of the composites.

**Figure 8 polymers-14-04946-f008:**
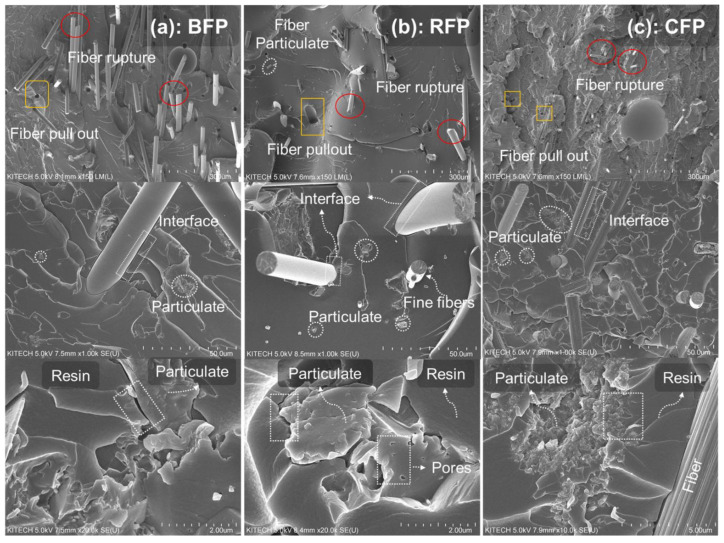
SEM images of the composites (**a**) Basalt composites, (**b**) Rockwool composites, and (**c**) Carbon composites. Fiber pull out, fiber interface, fiber particles, and particle interface are highlighted in the Figure.

**Table 1 polymers-14-04946-t001:** Physical, mechanical, and thermal properties of fibers are included in the study [40,41].

Properties	Carbon Fiber	Basalt Fiber	Rockwool Fiber
Filament diameter (µm)	5–15	6–21	1–15
Tensile strength of single filament (MPa)	3500–6000	3000–4840	490–770
Elastic modulus (GPa)	230–600	93–110	61.4
Elongation at break (%)	1.5–2.0	3.1–6	0.6
Specific gravity (g/cm^3^)	1.75–1.95	2.65–2.8	2.7
Melting temperature (°C)	3652–3697	1450	997
Thermal conductivity (W/m K)	5.28 ± 0.42	0.031–0.038	0.03
Price ($/kg)	30	7	3

**Table 2 polymers-14-04946-t002:** Abbreviations of the composite samples.

Sr. #	Samples	Abbreviations
1	Rockwool fibers	RF
2	Rockwool fibers & particulates	RFP
3	Basalt fibers	BF
4	Basalt fibers & particulates	BFP
5	Carbon fibers	CF
6	Carbon fibers & particulates	CFP

## Data Availability

Not applicable.

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
