# Peer review of "Exploring the Potential of Alternate Inorganic Fibers for Automotive Composites"

_polymers, 2022, doi:10.3390/polym14224946_

Round 1

Reviewer 1 Report

The paper studied 3 different kinds of reinforcements(used carbon fiber, basalt fiber, rockwool) for polymer matrix composites, and their mechanical and thermal properties were investigated and compared with each other. Howeverthere are some questions to be answered before consideration of publication.

1) The properties of fibers should be provided in the context, because the mechanical properties of composite is related to these.

2) As for application in automotive industry, What mechanical properties are the necessary?

3”Lower impact strength of BF and CF is related to lower stiffness, therefore CF composite had the lowest impact strength.” Is it correct?

Author Response

We are highly thankful for the precious time of reviewer spent in the review. We tried our best to respond each question carefully, in the attached response file. 

We hope, authors will now consider our current form suitable for publication. 

Reviewer 2 Report

Authors need to address following quires to improve the quality of the article.

1. Article must undergo language proof reading by the language experts or native speakers. As I have found grammatical errors

2. The abstract is a summary of the introduction, materials and method, results and conclusion. This order needs to be followed. The methodology, results (quantifying data) and conclusion component of the abstract should be properly captured.

3. Introduction doesn’t give the background of the study. It is advised improve the introduction significantly.

4. Literature review looks shallow, add quantitative results of literature.

5. Highlight contribution of the study to knowledge gap/specific problem.

6.  How and from where basalt fibers are procured?

7. Mention Span-to-depth ratio from flexural test

8. “Three (03) repetitions for each mechanical test of different composite laminates were performed, and average values of results were reported.” As per ASTM standards Five (5) repetitions need to be carried out. It is advised to follow the same.

9. “Material's coefficient of thermal expansion (CTE) CTE” remove repetitive word CTE.

10. “This could be related to poor dispersion and weak interface of rockwool fibers particles. As rockwool is an impure form of basalt, formed by the addition of clay to the basalt. This clay impurity renders brittle nature and lower tensile strength in rockwool fibers.” Corelate this statement with SEM images.

11. Results and discussion part looks shallow, add more reasonings and literature references.

12. “Comparing the results with and without  particulate inclusion, it can be observed that without particles the impact resistance is higher.” What is the reason for this?

13.” Figure 5(b) showed that all composite's inclusion of their particulates had a higher value of the loss modulus than their fiber-reinforced composites.” What is the reason for this?

14. SEM analysis is very poor. Need more evidence for the claims.

15. Kindly reconcile the conclusion with the study objectives.

16. What are the practical implications of this study and the future directions? kindly state

Author Response

(The authors gave the same response as above.)

Round 2

Reviewer 2 Report

Authors have addressed all the queries. Article may be accepted in present form.